# Dopaminergic Modulation of Prefrontal Cortex Inhibition

**DOI:** 10.3390/biomedicines11051276

**Published:** 2023-04-25

**Authors:** Danila Di Domenico, Lisa Mapelli

**Affiliations:** Department of Brain and Behavioral Sciences, University of Pavia, 27100 Pavia, Italy

**Keywords:** prefrontal cortex, dopaminergic system, GABAergic system

## Abstract

The prefrontal cortex is the highest stage of integration in the mammalian brain. Its functions vary greatly, from working memory to decision-making, and are primarily related to higher cognitive functions. This explains the considerable effort devoted to investigating this area, revealing the complex molecular, cellular, and network organization, and the essential role of various regulatory controls. In particular, the dopaminergic modulation and the impact of local interneurons activity are critical for prefrontal cortex functioning, controlling the excitatory/inhibitory balance and the overall network processing. Though often studied separately, the dopaminergic and GABAergic systems are deeply intertwined in influencing prefrontal network processing. This mini review will focus on the dopaminergic modulation of GABAergic inhibition, which plays a significant role in shaping prefrontal cortex activity.

## 1. The Prefrontal Cortex

The prefrontal cortex (PFC) is thought to be the highest association area in the mammalian cortex and is required for proper executive control. Task flexibility and planning [1], selective attention, attentional set-shifting, rule learning, strategy switching, and goal-directed behavior [2,3,4] are just some of the many PFC functions. This considered, it is not surprising that PFC alterations have been associated with a variety of psychiatric conditions. For example, several investigations reported PFC-related impaired working memory [5,6,7,8] and altered network oscillations [9,10] in schizophrenia. Though rodent PFC is less complex than that of primates, it exerts similar functions in the executive domain [11]. For this reason, the rodent represents a valuable model to investigate how PFC functions are determined at the molecular, cellular, and network levels. However, investigations in rodents are complicated by the lack of a univocal and unambiguous nomenclature of PFC subdivisions. Due to its recent evolution and inter-species variability, it is challenging to identify proper structural and functional criteria to define PFC regions [12,13]. This has been the subject of many studies aiming at characterizing differences and similarities of mammalian PFC [14]. Ref [15] introduced a hodological criterium based on the assumption that the mediodorsal thalamic nucleus (MD) is the primary site of projections toward the PFC. Therefore, according to this definition, the mammalian PFC could be identified based on the connectivity with the MD. Following this perspective, the effective existence in rats of two prefrontal cortex areas receiving projections from the MD, indicated as medial and orbitofrontal, was demonstrated [16]. Clearly, this definition bears some limitations. Indeed, other criteria were then adopted. For example, other researchers proposed a cytoarchitectural criterion, though this method was deemed valid only for closely related species [12]. To date, the best way to define PFC parcellation is proposed to be a combination of four criteria: function, architecture, connectivity, and topography [17,18]. In particular, the relevance of the connectivity aspect grew over time. Recent works have described the organization of cortical interconnectivity into modules along the whole brain [18,19] and identified a prefrontal cortical module. The areas within the prefrontal module show dense interconnections [20,21] and are believed to be devoted to similar functions [22]. The regions recognized as a component of the prefrontal module are the prelimbic area, the infralimbic area, the anterior cingulate area, the frontal pole cerebral cortex, and the orbital areas. Another widely used distinction, mainly based on connectivity mapping including thalamocortical, corticothalamic, corticostriatal, and corticocortical projections, recognizes three broad PFC subdivisions: the dorsomedial PFC (dmPFC), ventromedial PFC (vmPFC), and ventrolateral PFC (vlPFC). Considering the complex scenario of rodent PFC nomenclature and the absence of a standard reference for the different studies available in the literature, it is not surprising that many studies focusing on the PFC report vague indications of the subregion actually subjected to analysis. In particular, most investigations on the highest-level cognitive functioning in rodents target the so-called medial PFC (mPFC), comprising the infralimbic, prelimbic, and anterior cingulate areas [2,13]. It is worth specifying that there is no direct anatomical equivalence between human and rodent PFC. However, the rodent mPFC is anatomically located in correspondence with the anterior cingulate cortex (ACC) in humans (see [13] for a detailed review of the comparison between rodent and human PFC). Here, we will mainly refer to rodent reports on the mPFC, which is the most commonly addressed PFC area. The cytoarchitecture and the connectivity patterns are similar in rodents and humans, with the significant difference represented by the lack of the granular layer (layer IV) in rodent PFC. In both cases, the PFC is mainly composed of pyramidal neurons (PN, 80–90%) and inhibitory interneurons (IN, 10–20%) [23]. The main excitatory output is provided by the PNs, which are strongly interconnected to form a local network that projects to other cortical and subcortical areas. PN activity is modulated by a strong network of GABAergic INs [24,25], which proved to be essential for controlling PN firing and generating neuronal network oscillations [26,27,28]. The interplay between PNs and INs modulates PFC activity and is crucial to maintain proper cognitive functions.

## 2. Dopamine Receptors in the PFC

Dopamine (DA) is released in the mPFC by projections originating from the midbrain nuclei of the ventral tegmental area (VTA) and substantia nigra pars compacta [29,30]. Once released, DA interacts with five different receptors subtypes (D1, D2, D3, D4, D5) subdivided into two families: D1-like receptors comprising D1 and D5, and D2-like receptors comprising D2, D3, and D4 [29,31,32]. Receptors belonging to the D1-like family are more abundant than those of the D2-like family and are expressed in all PFC layers. On the other hand, receptors of the D2-like family are primarily expressed in deeper layers (mainly layer V) [33], and their affinity is 10–100 times higher than that of D1-like receptors [34]. Both DA receptor families are expressed on pyramidal and non-pyramidal neurons, thus modulating excitation and inhibition [29,33]. Finally, these two receptor classes differ in the intracellular signaling pathway mediating their effects. Since DA receptors are G-protein coupled receptors (GPCRs), they all activate heteromeric G-proteins, but the second messenger and the effector proteins activated are usually different for different receptors and, in most cases, mediate opposite responses.

In particular, D1-like receptors activation is coupled with the G-proteins Gα_s_ and Gα_olf_ which, in turn, are associated with adenylyl cyclase (AC) that, once activated, increases the level of cyclic adenosine monophosphate (cAMP) leading to the activation of protein kinase A (PKA). PKA modulates most D1-like functions by phosphorylating many substrates including voltage-gated K^+^, Na^+^, and Ca^2+^ channels, GABA receptors, and NMDA receptors [32,35]. One of the main PKA targets is the DA and cAMP-regulated phosphoprotein DARPP-32, which is crucial in regulating downstream signaling pathways. When phosphorylated, DARPP-32 inhibits the protein phosphatase 1 (PP1) that opposes PKA action, eventually amplifying PKA signaling. On the other hand, the activation of D2-like receptors leads to the opposite effect. When activated, these receptors couple with Gα_i_ and Gα_o_ that inhibit the activation of AC, thus limiting PKA signaling. Moreover, the activation of D2-like receptors determines the activation of the calmodulin-dependent protein phosphatase (PP2B), which turns DARPP-32 into a strong inhibitor of PKA signaling [32]. Thus, DARPP-32 can bidirectionally modulate PKA activity. Besides their regulation through PKA pathways, ion channels can also be modulated directly via binding the Gβγ subunit or indirectly via activation of the phospholipase C (PLC) by both D1-like and D2-like receptors (Figure 1). The latter is most common for modulating Ca^2+^ conductance, determining a decrease in Ca_V_2.2 (N-type) and Ca_V_1 (L-type) currents. PLC can also be activated through coupling with Gα_q_, though limited to cells expressing D5 and D1/D2 heterodimers [36,37]. Lastly, D1-like and D2-like receptors can modulate NMDA and GABA receptors through direct protein–protein interactions or PKA/IP3 signaling [35]. The mechanism by which D2-like receptors, particularly D4, regulate GABA receptors involves a pathway comprising the dephosphorylation of cofilin (an actin depolymerizing factor) via PP1 activation. This leads to the loss of actin stability, with a consequent interruption of myosin motor-mediated transport of GABA receptor-containing vesicles in the membrane, resulting in a reduced GABA receptor-mediated current [38].

## 3. Dopamine Modulation of GABAergic Inhibition

### 3.1. On Pyramidal Neurons (PN)

As nicely reviewed by [29], the net effect of DA release onto the PFC also depends on cell type, synaptic properties, and interactions with other neurotransmitters. One of the critical DA roles in the PFC is the modulation of the GABAergic system. This modulation contributes to setting the proper excitation/inhibition (E/I) balance in the PFC, which requires fine-tuning to ensure correct network activity. Indeed, the E/I ratio is disrupted in a broad range of psychiatric disorders [39,40,41]. Many studies focused on the role of D4 receptors in preserving the correct E/I balance. D4 receptors are enriched in the PFC and are usually expressed in dendritic processes [42,43,44], while D1 receptors are most prominent at PN dendritic spines [45]. In particular, D4 receptors are mainly expressed nearby GABA_A_ receptors in PFC PNs [46]. Experimental evidence showed that D2/D4 receptor agonists decrease the inhibitory post-synaptic currents (IPSCs) of layer V PN in rodent PFC, while a D1 receptor agonist increases IPSCs amplitude in the same neurons [46,47,48]. When D1- and D2-like receptors activation combines, an initial downregulation of the IPSCs mediated by D2-like receptors is followed by a D1-like receptors-dependent IPSCs increase. This suggests the biphasic nature of DA modulation of GABAergic responses in PFC PNs [29,47].

DA is reported to regulate inhibition through different intracellular mechanisms. In particular, high DA concentrations increase spontaneous inhibitory postsynaptic potentials (sIPSP) in PFC layer II/III [49] and layer V-VI PNs [50], revealing DA-mediated enhancement of GABA release. On the other hand, DA can depress evoked IPSP (eIPSP) in layer V-VI PNs [47,51,52]. This evidence shows that DA can modulate spontaneous and evoked IPSPs affecting GABA release mechanisms, hence regulating the presynaptic machinery [29]. This effect was also described in IN-PN pair recordings [53]. A possible explanation of the different DA impact on spontaneous and evoked IPSCs is proposed by [29]. The authors highlighted that the eIPSCs derive from activating a specific fiber through electrical stimulation, while sIPSCs derive from multiple diverse inputs. Therefore, the effect of DA on IPSCs may depend on the neuronal type generating the IPSC and the different neurons originating the GABAergic terminals impinging on that same neuron [29]. The heterogeneity of DA modulation reported in different studies might also depend on the recording sites. Indeed, D1- and D2-like receptors have different expression patterns: while D1-like receptors mRNA are also expressed in superficial layers, D2-like receptors are restricted to deeper layers such as layer V [33] (Figure 2).

### 3.2. On Inhibitory Interneurons (IN)

DA receptors are expressed in a wide array of GABAergic interneurons and, therefore, DA release onto the PFC affects IN activity, too [33,54,55]. DA is known to induce an increase in intrinsic excitability favoring depolarization in fast-spiking interneurons (FS) via a D1-like receptor-dependent mechanism [56,57]. Moreover, the effect of D1-like and D2-like receptors on PFC GABAergic INs may differ on a temporal scale. The activation of D1-like receptors induces both a depolarization and an increase in the neuronal excitability of FS. Different mechanisms mediate these two effects. The DA-induced depolarization lasts less than the increased excitability, meaning that DA can act through the same receptors to modulate different ionic currents at different time scales [56]. Interestingly, the activation of D2-like receptors at the peak of D1-like mediated IPSC determines a decrease in the IPSC amplitude [47,56]. Consistent with the biphasic hypothesis of DA modulation of the GABAergic system, D2-like receptors mediate a reduction in inhibition, and D1-like receptors mediate an increase in inhibition on PFC PNs, influencing IN activity (Figure 2). Lastly, D1-like receptors in superficial layers are often associated with vasoactive intestinal peptide (VIP) GABAergic INs and inhibit deeper INs via internal loops and interactions [58]. This supports the D1-like receptor role in determining circuit disinhibition, which is fundamental to appropriately modulating the PFC range of activity.

### 3.3. Evidence In Vivo

Several studies showed that DA exerts a predominantly inhibitory effect on PFC PN in vivo, primarily suppressing spontaneous firing [59,60,61]. Importantly, microdialysis data in vivo revealed a tonic level of DA in the PFC [62,63]. Most studies reported here were performed on anesthetized animals, where little VTA activity is presumably present at rest. Nevertheless, the stimulation of fiber bundles at the medial forebrain, or direct VTA stimulation, effectively increased DA levels in the PFC. It should also be considered that the absence of not experimentally evoked DA release is an advantage in characterizing transient DA effects on PFC neurons. For these reasons, these studies are considered suitable to address the consequences of DA release on the PFC in vivo. Indeed, VTA stimulation induces a fast EPSP-IPSP sequence in PFC PNs, with the IPSP consistent with GABA_A_ receptors activation [60]. Interestingly, the inhibitory component is eliminated not only by GABA_A_ receptor antagonists [64] but also by D2-like receptor antagonists, which tonically inhibit neuronal excitability [65,66,67]. When the D2-like receptor tone is abolished, the entire network physiology changes: neurons increase their firing, and the inhibition produced by VTA stimulation is occluded [29]. Overall, these studies show that DA released from dopaminergic terminals in the PFC, as well as exogenous DA, modulates spontaneous firing in vivo through complex mechanisms depending on the endogenous DA tone, the amount of DA released, and the activated receptor subtype. This effect was also confirmed by a computational model in which increasing DA concentrations elicited the facilitation of FS activity, with consequent suppression of pyramidal neurons firing. Moreover, enhancing basal DA levels rescues the initial condition, through the downregulation of the GABAergic tone, with consequent hyperactivity of PN firing [68]. Interestingly, computational models primarily based on in vivo studies have proposed a dual mechanism by which D1-like receptors can modulate working memory. First, the spontaneous activity of PN is decreased by upregulating inhibitory GABA currents; then, high-activity states are induced by upregulating excitatory NMDA currents [69,70]. This effect is believed to be mediated by D1-like receptors, which might induce inhibition by amplifying IPSCs in PNs [71], or an excitatory effect by enhancing NMDA receptor-mediated responses [72,73]. The same computational model was also used to implement D2-like receptors modulation of PFC activity. It was proposed that D2-like receptors activation decreases inhibitory currents in PNs while increasing IN excitability to maintain E/I balance [74].

Taken together, these findings provide evidence for a delicate homeostatic interplay between dopaminergic and GABAergic systems necessary to maintain PFC network stability and output selectivity.

### 3.4. Comments on PFC Regional Specificity

As pointed out in the first section, the PFC can be subdivided into several regions. It might then be of interest to consider whether a regional specificity has been observed in dopaminergic and GABAergic interplay. However, the intricate PFC subregions identification and nomenclature complicate the picture. Considering the literature reviewed here and mentioning to whatever extent the interaction between dopaminergic and GABAergic systems, it is not possible to infer a region specificity. Indeed, out of 24 studies, 14 reported to be generally on the mPFC (10) or PFC (4), 8 addressed the prelimbic or prelimbic/infralimbic region (without discrimination), and 2 specified the anterior cingulate cortex and the shoulder region or Fr2 region of the frontal cortex (without discrimination). Therefore, it is not possible to extrapolate differences in dopaminergic–GABAergic interaction among the mPFC subdivisions. Indeed, the prelimbic region seems to be the preferred target of most studies.

However, searching for a regional distinction might be pointless. Accumulating evidence suggests that current subdivisions might not reflect actual PFC functioning segregation. Recently, besides the cytoarchitecture and connectivity distinction criteria, the PFC gene expression profile has also been reported [75]. Interestingly, this study did not identify distinct subregions, but the genetic profile was, in fact, common to the multiple regions composing the PFC. Therefore, the PFC subdivisions based on connectivity or cytoarchitecture criteria, already not matching one another, are not confirmed by gene expression. This is of particular interest since it highlights a crucial aspect when considering PFC functions. The scenario that is emerging suggests that assigning different functions to the different PFC subdivisions is indeed deeply misleading. Based on connectivity alone, some distinctions seem to emerge, at least among the three main subdivisions (dmPFC, vmPFC, and vlPFC), which show different densities of specific connections. Nevertheless, their connectivity is not entirely differentiated, and the connections are shared but differ quantitatively [18]. Further based on this evidence, the dmPFC is often studied for sensorimotor behavior, the vmPFC is often associated with emotions and memory, while the vlPFC, though much less studied than the other two subdivisions, is often correlated to reward-related information and addictive behavior. However, this might reflect the common practice of the researchers rather than actual functional segregation. Several behavioral studies suggested that the perturbation of any PFC subdivision is sufficient to disrupt behavior and the whole cortical activity, independent of the type of task at hand (see [18] for an extensive discussion on this topic). Therefore, despite the different supposed roles of each subdivision, it is most likely that the PFC processes higher cognitive functions as a whole and cannot be assigned to a specific subregion [76].

## 4. Clinical Relevance

Given the evidence summarized so far, it is not surprising that several PFC-related pathologies involve alterations in both the dopaminergic and GABAergic systems. In the following paragraphs, we will briefly summarize the involvement of the dopaminergic and GABAergic systems in the main pathologies with a prominent PFC component, in particular schizophrenia and autism spectrum disorders.

**Schizophrenia** is one of the most studied cognitive pathologies, with a renowned involvement of the dopaminergic system, which is responsible for maintaining the proper E/I balance [77]. The “revised dopamine hypothesis” proposes that schizophrenic patients have hyperactive dopamine transmission in mesolimbic areas and hypoactive dopamine transmission in PFC [78]. The positive symptoms of schizophrenia include hallucinations and delusion due to an augmented dopamine release in subcortical areas, leading to an increase in D2-like receptors activation [79], and are thought to be caused by disrupted cortical pathways through the nucleus accumbens [80]. On the other hand, negative symptoms, such as anhedonia, lack of motivation, and speech impairments, result from reduced D1-like receptors activation in the PFC [79]. As computational models highlighted, the imbalance between D1-like and D2-like receptor activity might explain the positive and negative symptoms and the cognitive alterations in schizophrenia [81]. Interestingly, besides other players recently found involved (as the glutamatergic system and the NMDA receptors, [82,83]), the GABAergic system has been reported to be altered. In particular, a reduction in GABAergic inhibition is often reported (e.g., a reduced expression of GAD67, GAT1, and GABA_A_ receptors; a decreased number of inhibitory interneurons; reduced inhibitory currents; [84] for details). The investigations on GABAergic disruption in schizophrenia are complicated since the alterations differ depending on the specific targeted PFC region [85]. In any case, GABAergic signaling alterations will contribute to the E/I balance disruption associated with this disease, both in humans and animal models. Alterations in GABA release have been correlated with impaired gamma oscillations and, as such, to the cognitive symptoms of the disease [86]. Interestingly, the GABAergic system deficit in the PFC has been proposed to result from the altered dopaminergic tone in the striatum in a mouse model with striatal D2 receptors overexpression [87]. Though the idea that the GABAergic and dopaminergic systems influence each other and collaborate in determining the pathological alterations in schizophrenia is not new [88,89], further research on this interaction might reveal critical to disentangle the complex pathophysiology of the disease. This would have a relevant impact from the clinical perspective. Independent of where the primary alteration occurred, a clinical intervention might need to impact both systems to regain a proper balance in PFC network activity. Moreover, a complete view of such a complex pathology will need to integrate the alterations seen in other neurotransmitter systems (such as the glutamatergic one) and the impact on the E/I balance of the glutamate/GABA interplay [77].

**Autism spectrum disorders** (ASD) are neurodevelopmental disorders characterized by deficits in social cognition, repetitive and stereotyped behavior, and restricted interests. The investigation of the pathophysiology of ASD is complicated by the incredibly heterogeneous genetic and phenotypic profiles that can be found in humans and the several animal models of the disease [90,91]. Nevertheless, the diverse molecular, cellular, and network alterations reported in literature seem to converge on a common outcome characterized by altered E/I balance (in favor of excitation), network hyperexcitability, and hyperresponsivity, often accompanied by altered long-range connectivity [92,93,94]. The GABAergic system is considered central for ASD research, and its interplay with the glutamatergic one to determine the E/I balance is one of the most studied topics in this field [95]. The most common alteration reported is a decrease inhibition efficiency, ultimately leading to the complex cognitive dysfunctions reported and the comorbidity with anxiety and other disorders [84]. The involvement of the dopaminergic system in ASD is supported by significant evidence in humans and animal models [96] and confirmed by the contribution of alterations in genes related to DA neurotransmission and its modulation [97]. The prefrontal cortex and striatum are considered the most affected brain regions. Given the role of the dopaminergic system in fine-tuning network transmission and signal-to-noise ratio during behavior, alterations in this system are considered causal for the reduced sociability and increased repetitive behavior that characterize ASD phenotype in mice and, most likely, in humans [98,99]. Therefore, ASD physiopathology could be the ideal ground to study the correlation between DA and GABAergic system alterations.

**Affective disorders**, such as major depression and bipolar disorder, and **anxiety disorders** are commonly associated with altered serotoninergic tone and glutamate/GABA systems imbalance. Nevertheless, many symptoms are considered to rely on dopaminergic miscontrol leading, for example, to a lack of motivation and anhedonia in depression [100,101]. In particular, many forms of depression have been correlated with PFC hyperactivity, and acting on the systems controlling the E/I balance in this region is the primary treatment approach to date [100]. The circuits responsible for the stress response, including the hippocampus and amygdala, are also involved in the altered PFC-related communication found in these disorders [100,102]. Altered DA signaling is also reported in post-traumatic stress disorder [103]. Moreover, the involvement of the dopaminergic system in pain modulation and **chronic pain** can be considered related to the previous disorders [104,105]. Interestingly, the increased mPFC output observed in neuropathic pain conditions has been correlated with altered VTA-mediated DA control over the prelimbic region in rats, associated with impaired integration of GABAergic inhibition [106].

## 5. Conclusions

The dopaminergic system modulates the PFC network activity state, finely tuning the signal-to-noise ratio and the E/I balance. These effects are partially exerted influencing the GABAergic system through complex intracellular pathways that modify GABA receptors expression and activity, and modulate GABA release by INs. DA control of the PFC activity state and responsiveness modulates the gain of signal transmission modifying the tonic DA level and regulates the timing of neuronal responses through its complex phasic component. The PFC is one of the most integrative areas in the brain, and the interplay between the dopaminergic and GABAergic systems is one of the critical features that influence input integration by this network and therefore deserves special attention. Further effort should also be devoted to exploring the reciprocal influence of these two systems in PFC-related neuropathologies. More often than not, the alterations in DA and GABAergic systems and their impact on the clinical perspective are studied separately. This is undoubtedly due to the intrinsic difficulty in disentangling the relative contribution of the two systems to the alterations observed and to the limitations of using animal models for addressing cognitive phenotype. Nevertheless, the data summarized in this mini review strongly support the idea that the interplay between these two systems significantly contributes to originate the unbalance seen in pathological models, possibly with a primarily affected system causing the impairment of the other. The recent technological advancements and the application of computational models could boost the research in this field and allow us to address this issue with a renewed effort.

## Figures and Tables

**Figure 1 biomedicines-11-01276-f001:**
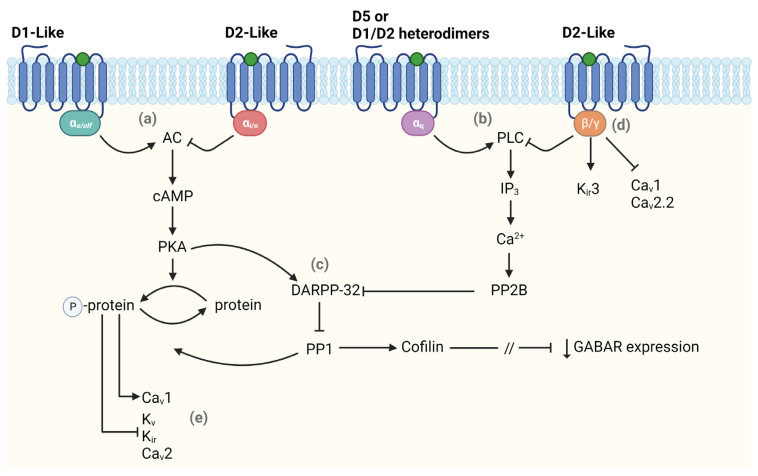
Main intracellular pathways activated by dopamine receptors. The scheme shows different pathways in which dopamine (DA) affects the modulation of intracellular signaling. DA can regulate the activation state of (**a**) adenylyl cyclase (AC) or (**b**) phospholipase C (PLC) binding either D1-like or D2-like receptors. (**c**) Both pathways lead to a modulation (either positive or negative) of DARPP-32 which regulates the expression of GABA receptors. DA also affects neuronal excitability by modulating voltage-dependent ion channels via activation of (**d**) β/γ subunit or (**e**) AC pathway. The forward and stop arrows indicate activation or inhibition of the next element in the chain, respectively. This figure was created with BioRender.com.

**Figure 2 biomedicines-11-01276-f002:**
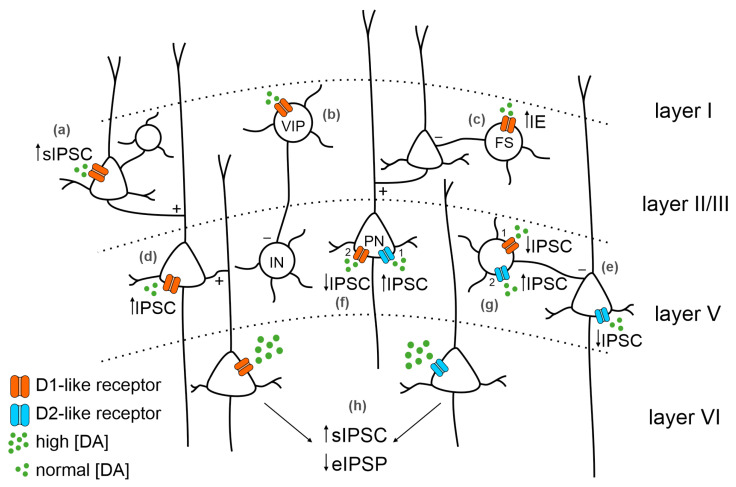
Dopaminergic receptors distribution in the PFC and main effects on inhibition. The distribution of dopamine (DA) receptors among PFC layers and their expression on different neuronal types can variably affect inhibition. In layer II/III, DA (green dots) binding D1-like receptors (orange) on pyramidal neurons (PNs, (**a**)) increases spontaneous IPSC (sIPSC); (**b**) on vasoactive intestinal peptide (VIP) neurons, it starts internal loops inhibiting deeper layers’ inhibitory interneurons (INs); and (**c**) on fast-spiking interneurons (FS) increases intrinsic excitability. DA binding D1-like receptor expressed in layer V PNs (**d**) increases the IPSC. DA binding D2-like receptors (blue) expressed in layer V PNs (**e**) decreases the IPSC. Expression of both D1-like and D2-like receptors in layer V PNs (**f**) increases the IPSC mediated by D2-like receptor activation (1) followed by a IPSC decrease mediated by D1-like receptor activation (2). On INs (**g**), the decrease in the IPSC mediated by D1-like receptors (1) is followed by an increased IPSC mediated by D2-like receptors (2). (**h**) In layer VI PNs, the activation of DA receptors by high DA concentration leads to an increase in sIPSC and a decrease in evoked IPSC (eIPSC).

## Data Availability

No new data were created or analyzed in this study. Data sharing is not applicable to this article.

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
