# Peer review of "Dopaminergic Modulation of Prefrontal Cortex Inhibition"

_biomedicines, 2023, doi:10.3390/biomedicines11051276_

Round 1
Reviewer 1 Report
The mesocorticolimbic system is thought to be involved in a variety of processes that require deliberation and decision making. In addition to its normal functions, dysregulation of this system is associated with drug abuse and several psychiatric disorders. Anatomically, the focus of research has been on the ventral tegmental projections to the nucleus accumbens and prefrontal cortex. The prefrontal cortex has received less attention than the ventral tegmentum or nucleus accumbens despite a perhaps having a pinnacle role in this circuit. Further, because the output neurons in this system predominate, these neurons have received primary experimental attention. At the same time, critical roles for interneurons at each level of the circuit have been identified. This is a well-written review that highlights the GABA interneurons in the prefrontal cortex and their interactions with dopamine signaling in the projection neurons. The functional roles for these interactions, especially in terms of dysregulation, are explored. This is a valuable review which will be well-received by both experts and non-experts interested in prefrontal cortical function.
My only concern with the review is that (as the authors note) there are different subdivisions of the prefrontal cortex which have different anatomical targets. The scope of the review could be expanded by examining whether GABA-dopamine interactions differ among these prefrontal cortical subdivisions.
Author Response
We thank the Reviewer for the positive comments and for raising an interesting point. Indeed, the matter of prefrontal cortex (PFC) subdivisions is quite debated. There is little consensus about the criteria used for identifying different regions, both from the anatomical and functional perspective. Therefore, at first, we expanded the introductory section on the PFC (lines 28-62 of the revised version of the manuscript) explaining the complex issue of subdivisions. Then, we further analyzed the literature used for our review and observed that “out of 24 studies, 14 reported to be generally on the mPFC (10) or PFC (4), 8 addressed the prelimbic or prelimbic/infralimbic region (without discrimination), 2 specified the anterior cingulate cortex and the shoulder region or Fr2 region of the frontal cortex (without dis-crimination). Therefore, it is not possible to extrapolate differences in dopaminergic-GABAergic interaction among the mPFC subdivisions”, as now reported in the text.
To account for this useful specification, we added an entire new section on this matter: “3.3. Comments on PFC regional specificity” (lines 228-263).
We believe that addressing this issue significantly improved our manuscript.
Reviewer 2 Report
The authors have done an excellent job with the assessment of literature pertaining to this field of research.
The manuscript attempts to address the current understanding of the role of dopaminergic and GABAergic systems and their interplay in the prefrontal cortex in health. The connection between these two systems that are responsible for excitatory (motivational) and inhibitory influences respectively, especially using the local interneurons provides a good review of how these processes dictate the executive function of the central nervous system.
The review is a compilation of recent research to provide an adequate introduction to the reader about the necessity of the interplay for the function of the frontal cortex executive function.
I have not seen many other reviews on this topic in this journal.
The review provides a point of view to look at how dopaminergic and GABAergic system interact in the interneurons of the frontal cortex to effect executive control over synaptic function in the central nervous system. The conclusions are commensurate with the scope of the work presented and the current view of the literature available.
References - Adequate for the subject matter of the review.
The figures provide a pictorial view of the subject matter discussed – there are creative suggestions in terms of placement of the captions and improvements in representation that can emphasize the relevant aspects, but it is a matter of preference and does not take away from the material being adequate for the scope of the publication.
Author Response
We thank the Reviewer for the positive comments and for critically addressing our paper. Following your suggestions, we tried to improve the figures to emphasize the most relevant aspects. However, we did not find a better representation, since in any case we would lose readability of other relevant aspects.
Reviewer 3 Report
Revision biomedicines-2349160 – “Dopaminergic modulation of prefrontal cortex inhibition”.
This short literature review by Domenico et al. is designed to review the most relevant research that highlight the impact dopaminergic maladaptive signaling in PFC inhibitory networks.
This review was a pleasure to read, and the most relevant achievements to the topic are well-discussed. Per my comment above – the manuscript needs only minor corrections:
(1) The manuscript would benefit if the authors include a section for abbreviations.
(2) Please change the references insertion throughout the manuscript using the standards of the journal. For example: line 22, please change [2]-[4] to [2-4]; and line 25, change [9],[10] to [9, 10].
(3) Please use a uniform designation throughout the manuscript: (page 3, line 103), “GABAA receptors”; and (page 6, line 216), “GABA-A receptors”.
Author Response
We thank the Reviewer for the extremely positive comments, we are glad that our work was appreciated.
We addressed the minor concerns that were raised, as specified below.
1) A list of abbreviations was added at the end of the manuscript.
2) We changed the reference style in order to completely match the Journal standard.
3) We checked the text for the use of uniform abbreviations and corrected when needed.